# Preoperative Home-Based Multimodal Physiotherapy in Patients Scheduled for a Knee Arthroplasty Who Catastrophize About Their Pain: A Randomized Controlled Trial

**DOI:** 10.3390/jcm14010268

**Published:** 2025-01-05

**Authors:** Marc Terradas-Monllor, Hector Beltran-Alacreu, Mirari Ochandorena-Acha, Ester Garcia-Oltra, Francisco Aliaga-Orduña, José Hernández-Hermoso

**Affiliations:** 1Research Group on Methodology, Methods, Models and Outcomes of Health and Social Sciences (M3O), Faculty of Health Sciences and Welfare, Centre for Health and Social Care Research (CESS), University of Vic-Central University of Catalonia (UVic-UCC), 08500 Vic, Spain; marc.terradas@uvic.cat; 2Institute for Research and Innovation in Life Sciences and Health in Central Catalonia (IRIS-CC), 08500 Vic, Spain; 3Toledo Physiotherapy Research Group (GIFTO), Faculty of Physical Therapy and Nursing, Universidad de Castilla-La Mancha, 45071 Toledo, Spain; hector.beltran@uclm.es; 4Orthopedic Surgery Department, Germans Trias i Pujol University Hospital, 08916 Badalona, Spainjahernandezh.germanstrias@gencat.cat (J.H.-H.); 5Department of Surgery, Faculty of Medicine, Universitat Autònoma Barcelona, 08193 Cerdanyola del Vallès, Spain

**Keywords:** knee osteoarthritis, total knee arthroplasty, physiotherapy, chronic postsurgical pain, pain catastrophizing

## Abstract

**Background**: Chronic pain affects about 20% of total knee arthroplasty (TKA) patients, with high pain catastrophizing being a key predictor. Screening and addressing this modifiable factor may improve postoperative outcomes. **Objective**: We aimed to compare the effectiveness of two preoperative home-based multimodal physical therapy interventions on pain catastrophizing in high-catastrophizing TKA patients. Secondarily, the study aimed to assess postoperative outcomes over six months. **Methods**: A total of 40 patients with symptomatic osteoarthritis and moderate pain catastrophizing were randomly allocated to the control, therapeutic patient education (TPE), and multimodal physiotherapy (MPT) groups. Preoperative interventions comprised pain neuroscience education, coping skills training, and therapeutic exercise, differing in the number of sessions and degree of supervision. All outcomes were assessed before and after the treatment in the preoperative period, and 1, 3, and 6 months post-surgery. The primary outcome measure was pain catastrophizing. **Results**: Both intervention groups showed a preoperative reduction in pain catastrophizing. TPE patients had lower pain ratings at rest and lower catastrophizing scores at 1 and 6 months post-surgery, reduced kinesiophobia and improved dynamic balance at 3 and 6 months post-surgery, and higher self-efficacy at 1 month post-surgery. MPT patients exhibited lower pain catastrophizing and pain intensity during walking at 1 month post-surgery, and better outcomes in kinesiophobia, self-efficacy, and dynamic balance at 1, 3, and 6 months post-surgery, along with higher walking speed at 6 months post-surgery. **Conclusions**: Preoperative physiotherapy reduces preoperative pain catastrophizing and improves postoperative pain-related outcomes, behaviors, and cognitions in high-catastrophizing TKA patients. Registration is with the United States Clinical Trials Registry (NCT03847324).

## 1. Introduction

Chronic postsurgical pain is pain that develops or increases in intensity after a surgical procedure or a tissue injury and persists beyond the healing process [1]. The prevalence of chronic pain after total knee arthroplasty (TKA) is usually set at around 20% of the patients [2]. Among the most significant risk factors, pain catastrophizing and pain intensity emerged as two of the strongest predictors for poor postoperative outcomes, including disability [3,4], poor quality of life [5], and chronic pain [4,6,7,8,9]. Pain catastrophizing has been conceptualized as the tendency to magnify the threat value of a pain stimulus, and to feel helpless in the context of pain, and as a relative inability to inhibit pain-related thoughts in the anticipation of, during, or following a painful event [10]. However, a recent study by Petrini and Arendt-Nielsen proposed a re-conceptualization of this construct within a modern biopsychosocial framework. They described it as an emotional regulation strategy and a form of catastrophic worry (characterized by repetitive negative thinking), arising from the interplay between a ruminative process and personality trait characteristics associated with the behavioral inhibition system [11]. Numerous authors suggest that screening for high-catastrophizing subjects and delivering interventions designed to reduce pain catastrophizing and pain intensity before surgery may have the potential to further improve joint replacement outcomes [12,13,14]. Few studies have tested this hypothesis; Birch et al. [15] and Buvanendran et al. [16] assessed the effectiveness of a cognitive behavioral therapy intervention on subjects scheduled for a TKA and high pain catastrophizing. At the same time, Riddle et al. [17] examined the influence of pain-coping skills training in this population. No differences were observed between groups in any of the previous studies. However, pain catastrophizing reduction before surgery was only reported in one of the previous studies [16].

Although cognitive behavioral therapy appears to have the best evidence in these cohorts [18], multimodal physical therapy has been shown to effectively reduce pain intensity and pain catastrophizing in symptomatic osteoarthritis patients scheduled for a TKA who catastrophize about their pain [19]. Therefore, the primary aim of this study is to compare the effectiveness of two preoperative, home-based, multimodal physical therapy interventions—both incorporating pain neuroscience education, therapeutic exercise, and coping skills training—in reducing preoperative pain catastrophizing in individuals with high pain catastrophizing who are scheduled for a total knee arthroplasty. Secondarily, the preoperative and postoperative effects on pain catastrophizing, health functioning, pain-related fear of movement, self-efficacy, walking speed, and dynamic balance were also investigated.

## 2. Materials and Methods

### 2.1. Design

A single-blinded, randomized controlled trial with three parallel groups was conducted between September 2019 and January 2023 at the Hospital Germans Trias I Pujol. The study was registered in the United States Clinical Trials Registry (NCT03847324). The study was approved by the Human Research Ethics Committee of the Hospital Germans Trias i Pujol (PI-18-239), and the Research Ethics Committee of the University of Vic–Central University of Catalonia (58/2018). This research is reported according to the Consolidated Standards of Reporting Trials (CONSORT) Statement for Randomized Trials of Nonpharmacologic Treatments [20].

### 2.2. Participants

A previous feasibility study describes the participants’ inclusion criteria and methods in detail [19]. Participants were eligible for enrollment if they (i) were scheduled for TKA due to symptomatic primary osteoarthritis; (ii) were non-responsive to non-operative treatment; (iii) had moderate to severe knee pain (40 mm on a 100 mm visual analog scale) [21]; (iv) had a minimum score of 20 points on the Pain Catastrophizing Scale (PCS) [22]; (v) were willing to participate in the study; (vi) were able to read and understand the Spanish language. Participants were excluded if they (i) were scheduled for revision surgery or a unicompartmental arthroplasty; (ii) had a diagnosis of inflammatory arthritis. Orthopedic surgeons (EGA, FAO, and JHH) assessed participants for inclusion once they were scheduled for TKA.

### 2.3. Sample Size Calculation

The sample estimation was calculated through a previous feasibility study [19]. We used the software G*power3 (version 3.1, Düsseldorf, Germany) and the PCS as a primary outcome measure [23]. The effect size (η^2^) of the intervention groups relative to the control group was calculated for the post-treatment PCS score. With an η^2^ = 0.37, an effect size f = 0.78, a power of 0.90, an α level of 0.05, and a dropout rate of 25%, it was estimated that 33 participants would be required (11 per group).

### 2.4. Randomization and Blinding

The participants were randomly allocated to the control and intervention groups using a randomization schedule using a web-based random number generator (GraphPad QuickCalcs (Boston, MA, USA)) and numbered, opaque sealed envelopes [24]. Only the physiotherapist who performed the assessments was blinded.

### 2.5. Interventions

For a detailed description of the interventions, see the following feasibility study: [19]. Participants allocated to the control group received usual care, which included a multidisciplinary, group-based, biomedical preoperative education session and early inpatient physiotherapy focused on early mobilization and ambulation. This phase began on the day of the surgical intervention and continued until discharge, typically around four days post-surgery. Following discharge, each patient was assigned a physiotherapist through an external domiciliary rehabilitation service, which generally started within one to ten days after discharge. This was followed by home-based postoperative physiotherapy.

In addition to usual care, the therapeutic patient education group (TPE) received three home-based, one-on-one sessions of preoperative therapeutic education, composed of pain neuroscience education (PNE) and coping skills training (CST). PNE is an innovative intervention that aims to enhance the patient’s understanding of the biological processes within the pain construct, to improve the patients’ beliefs and coping strategies. In addition, CST aimed to train the patients in therapeutic exercise, self-mobilizations, and relaxation techniques. Both elements of this intervention, PNE and CST, were designed to improve patients’ pain and function and reduce their maladaptive cognitions, such as pain catastrophizing.

The multimodal physiotherapy group (MPT) received eight home-based, one-on-one sessions of preoperative multimodal physiotherapy. The intervention consisted of therapeutic education, supervised therapeutic exercise, and orthopedic manual therapy. Therapeutic education was based on PNE and CST, as previously described. Patients in this group received more intensive supervision during exercise training, with increased monitoring of exercise compliance and progression over an 8-week period. The therapeutic exercise program targeted strengthening, balance, and stability, and was organized into exercises focused on the core, hip, knee, and global regions. Core exercises included selective abdominal activation, kneeling abdominal planks, and kneeling side planks. For the hip and pelvic regions, participants performed supine pelvic lifts, one-legged pelvic lifts, lateral hip abductions, and standing hip abductions using elastic bands. Knee-focused exercises included isometric quadriceps contractions, knee extensions with elastic bands, weighted straight leg lifts, half squats with wall support, and alternating leg lunges. Finally, global exercises included one-legged standing, weighted step-ups, slow step-downs, transitioning from a supine to sitting position, and diaphragmatic breathing as a relaxation technique. Orthopedic manual therapy was another key component of this intervention, aimed at alleviating pain and improving mobility. Specific techniques were applied based on the identification of dysfunctions during the intervention period. If participants exhibited a loss of knee range of motion, whether in extension or flexion, mobilizations throughout the range of motion, end-range mobilizations, and mobilizations with movement were performed. Patellar mobility was assessed and mobilized as needed. For muscular tension, dynamic soft tissue mobilizations were employed, while nerve mobilizations and knee tractions were routinely used to modulate pain.

The intervention began 2–3 months before surgery during the waiting list period, and participants were encouraged to adhere to the physiotherapist’s recommendations regarding coping skills and therapeutic exercises. Compliance with the intervention was assessed at the post-treatment follow-up. After surgery, all participants transitioned to the standard care protocol.

### 2.6. Outcome Measures

One blinded physiotherapist performed each outcome assessment at baseline, at the end of the intervention, 8 weeks after the baseline assessment point, and 1, 3, and 6 months after surgery. The assessor underwent training prior to the study to ensure strict adherence to standardized protocols. Although most variables were collected using self-reported questionnaires, all items were thoroughly reviewed, and standardized procedures were implemented to clarify participants’ questions and provide feedback consistently. The assessments were conducted by a single physiotherapist with extensive experience in these procedures, ensuring uniformity and eliminating variability between assessors. Additionally, periodic reviews of assessment procedures were performed to maintain quality control and ensure compliance with the established protocols. The primary outcome measure was pain catastrophizing. Pain catastrophizing was assessed using the Spanish version of the Pain Catastrophizing Scale (PCS), a 13-item self-administered questionnaire with three subscales: rumination, magnification, and helplessness [22]. Secondary outcome measures included pain intensity, which was assessed using a 100 mm visual analog scale (VAS) [0 = no pain, 100 = worst imaginable pain] while resting and walking [25]. Health functioning used the Western Ontario and McMaster University Index (WOMAC), a multidimensional scale with 24 items grouped into three dimensions, pain, stiffness, and physical function, using a 5-point Likert scale [0 = none, 4 = extreme] [26]. Pain-related fear of movement was measured using the Tampa Scale of Kinesiophobia-11 (TSK-11), an 11-item self-administered questionnaire, with scores ranging from 11 to 44, higher scores indicating greater fear of movement [27]. Self-efficacy was measured using the Chronic Pain Self-Efficacy Scale (CPSES), a 19-item self-administered questionnaire using a 10-point Likert scale [0 = totally incapable, 10 = totally capable] [28]. Walking speed was assessed using the 4 m Walking Test (4mWT), where patients walked as fast as possible along an 8 m path, with 2 m for acceleration and deceleration [29]. And dynamic balance was assessed using the Y-Balance Test (YBT), which requires a single-limb stance on an elevated central footplate, pushing a rectangular reach indicator block with the non-standing foot in three directions (anterior, medial, lateral) [30]. YBT was not assessed at the 1-month follow-up for safety reasons.

### 2.7. Statistical Analysis

The Statistical Package for the Social Sciences (SPSS 28, SPSS Inc., Chicago, IL, USA) was used for the statistical analysis. Descriptive statistics were used to summarize baseline data, including means and standard deviations for continuous variables and numbers and percentages for categorical variables. A normal distribution of all measures was assessed using the Shapiro–Wilk test (*p* > 0.05), and due to the lack of normal distribution, the authors used nonparametric statistics. Continuous variables were compared using the Kruskal–Wallis test, while categorical variables were compared by the chi-squared or linear chi-squared test, depending on the number of categories. For between-group comparisons, the Kruskal–Wallis test was used. In the case of a significant Kruskal–Wallis test, a pairwise Dunn’s test was computed for post hoc between-group comparisons. Significance values were adjusted using a Bonferroni correction. Friedmann’s ANOVA was computed for intragroup comparisons, and the Wilcoxon signed rank test was calculated for the preoperative, perioperative, and postoperative periods. Finally, using the Kruskal–Wallis H-test statistic, the epsilon-squared ($\varepsilon_{R}^{2}$) effect size estimate was calculated using the following formula: $\varepsilon_{R}^{2}\frac{H}{\left(n^{2}-1)/(n+1\right)}$, where n is the number of participants [31]. The values of epsilon squared were considered very small when <0.01, small between 0.01 and <0.06, medium between 0.06 and <0.14, and large if 0.14 or higher [32]. Moreover, using the Wilcoxon Z-test statistic, the effect size r was estimated using the following formula: $r=\frac{Z}{\sqrt{n}}$ [31]. The r values were considered small when they were higher than 0.1, medium when higher than 0.3, and large when higher than 0.5 [32].

## 3. Results

Between 19 September 2019 and 18 March 2022, 126 patients were assessed for eligibility but only 40 met the inclusion criteria and were randomly allocated to either the control (n = 15), the TPE (n = 13), or the MPT (n = 12) treatment groups. A total of 86 patients were excluded. In the control group, two participants were lost due to the COVID-19 lockdown, and one was excluded because the patient did not receive the surgery. Three participants were also lost in the TPE group due to the COVID-19 lockdown. Furthermore, two participants were lost in the MPT group for the same reason. Details of exclusion reasons and follow-up data are shown in Figure 1.

The baseline demographic and clinical characteristics of the participants are shown in Table 1. No differences were observed between groups at baseline.

### 3.1. Between-Group Comparisons (Table 2)

At baseline (T0), no statistically significant differences were discerned across any of the outcome measures. However, at the post-treatment follow-up (T1), notable statistical differences emerged among the groups in several metrics, including PCS, VAS Walk, WOMAC total score, TSK-11, CPSES, YBT-A, YBT-M, and YBT-L. Specifically, the TPE group exhibited superior outcomes compared to the control group in PCS, VAS Walk, TSK-11, and YBT-A. Furthermore, the MPT group outperformed the control group in PCS, WOMAC total score, TSK-11, CPSES, YBT-A, YBT-M, and YBT-L. One-month post-surgery (T2), significant differences persisted among the groups in PCS, VAS Rest, VAS Walk, TSK-11, and CPSES. The TPE group continued to demonstrate superior performance relative to the control group in PCS, VAS Rest, and CPSES. Similarly, the MPT group exhibited better outcomes than the control group in PCS, VAS Walk, TSK-11, and CPSES. At the three-month post-surgery follow-up (T3), only TSK-11, CPSES, YBT-A, and YBT-M manifested statistically significant inter-group differences. The TPE group surpassed the control group in TSK-11 and YBT-A, while the MPT group excelled in TSK-11, CPSES, and YBT-A. Lastly, at the six-month post-surgery follow-up (T4), significant disparities were observed in PCS, VAS Rest, TSK-11, CPSES, 4 m Walking Test, YBT-A, YBT-M, and YBT-L. The TPE group outperformed the control group in PCS, VAS Rest, TSK-11, and YBT-M. Concurrently, the MPT group demonstrated superior outcomes compared to the control group in TSK-11, CPSES, 4 m Walking Test, and YBT-L. A graphical representation of the results can be found in the Appendix A.

Effect size estimation showed that, at T1, a large effect size was observed in PCS, VAS Rest, VAS Walk, WOMAC total score, TSK-11, CPSES, YBT-A, YBT-M, and YBT-L. At T2, a large effect size was observed in PC, VAS Rest, VAS Walk, WOMAC total score, TSK-11, and CPSES. At T3, a large effect size was displayed for TSK-11, CPSES, YBT-A, and YBT-M. Furthermore, at T4, a large effect size was observed in PCS, VAS Rest, WOMAC total score, TSK-11, CPSES, 4 m Walking Test, YBT-A, YBT-M, and YBT-L.

**Table 2 jcm-14-00268-t002:** Between-group comparisons.

**Variables**	**Groups**	**Baseline (T0)**		Post-Treatment (T1)		1 Month Post-Surgery (T2)		3 Months Post-Surgery (T3)		6 Months Post-Surgery (T4)		$\mathrm{Kruskal}\text{–}\mathrm{Wallis}\;\mathrm{Test}\;(p\text{-}\mathrm{Value});\;\mathrm{Effect}\;\mathrm{Size}\;(\varepsilon_{R}^{2})$
		**Median (IQR)**	**Dunn’s Test**	**Median (IQR)**	Dunn’s Test	Median (IQR)	Dunn’s Test	Median (IQR)	Dunn’s Test	Median (IQR)	Dunn’s Test	
PCS (0–52)	Con	36 (13)	-	37 (16)	0.024 ***** 0.002 ^†^ 1.000 ^‡^	23 (22)	0.018 ***** 0.046 ^†^ 1.000 ^‡^	8 (20)	-	12 (20)	0.007 ***** 0.194 ^†^ 0.688 ^‡^	0.121; 0.108 ^T0^ 0.001; 0.357 ^T1^ 0.010; 0.287 ^T2^ 0.173; 0.103 ^T3^ 0.008; 0.291 ^T4^
	TPE	28 (13)		19 (8)		6 (23)		2 (7)		0.5 (2)		
	MPT	28.5 (14)		13.5 (18)		11 (16)		3 (7)		2 (6)		
VAS Rest (0–10)	Con	6 (4)	-	5 (3.8)	-	5 (3.5)	0.038 ***** 0.672 ^†^ 0.729 ^‡^	2.5 (3.9)	-	1 (3.9)	0.032 ***** 0.060 ^†^ 1.000 ^‡^	0.485; 0.037 ^T0^ 0.065; 0.148 ^T1^ 0.044; 0.189 ^T2^ 0.260; 0.077 ^T3^ 0.015; 0.247 ^T6^
	TPE	5 (2.8)		4 (3.8)		3 (4)		0 (2)		0 (0)		
	MPT	5.5 (3.1)		3.75 (3.3)		4 (3)		2 (4)		0 (0)		
VAS Walk (0–10)	Con	8 (3)	-	8 (2)	0.046 ***** 0.357 ^†^ 1.000 ^‡^	6 (3.3)	0.067 ***** 0.049 ^†^ 1.000 ^‡^	3.5 (4.3)	-	2 (5.3)	-	0.128; 0.105 ^T0^ 0.048; 0.164 ^T1^ 0.022; 0.230 ^T2^ 0.228; 0.084 ^T3^ 0.388; 0.056 ^T4^
	TPE	9 (2)		6 (2)		3 (3)		1 (5)		1.5 (4.1)		
	MPT	8 (2.6)		7 (4.9)		3.5 (3.1)		2 (5)		1 (3)		
WOMAC Pain (0–20)	Con	11 (6)	-	11 (6)	-	7 (4)	-	5 (4)	-	4 (3)	-	0.601; 0.026 ^T0^ 0.128; 0.111 ^T1^ 0.347; 0.064 ^T2^ 0.588; 0.030 ^T3^ 0.173; 0.103 ^T4^
	TPE	12 (5)		8 (3)		7 (4)		4 (5)		2 (4)		
	MPT	10 (5)		8 (3)		6 (3)		3 (5)		2 (4)		
WOMAC total score (0–96)	Con	54 (25)	-	55 (24)	0.442 ***** 0.017 ^†^ 0.535 ^‡^	40 (21)	-	21 (15)	-	16.5 (20)	-	0.867; 0.007 ^T0^ 0.022; 0.207 ^T1^ 0.059; 0.172 ^T2^ 0.433; 0.048 ^T3^ 0.059; 0.167 ^T4^
	TPE	57 (25)		47 (10)		28 (22)		21 (19)		8 (10)		
	MPT	51 (22)		37 (18)		29 (18)		16 (23)		4 (14)		
TSK-11 (11–44)	Con	33 (8)	-	37 (8)	0.002 ***** 0.006 ^†^ 1.000 ^‡^	33 (5.25)	0.050 ***** 0.007 ^†^ 1.000 ^‡^	31 (5)	0.001 ***** 0.001 ^†^ 1.000 ^‡^	33 (7)	0.003 ***** 0.001 ^†^ 1.000 ^‡^	0.213; 0.079 ^T0^ <0.001; 0.390 ^T1^ 0.006; 0.325 ^T2^ <0.001; 0.505 ^T3^ <0.001; 0.496 ^T4^
	TPE	31 (8)		25 (8)		25 (17)		24 (9)		23 (10)		
	MPT	30 (18)		27.5 (12)		22.5 (8)		22 (10)		22 (13)		
CPSES (0–190)	Con	78 (60)	-	78 (47)	0.067 ***** 0.001 ^†^ 0.181 ^‡^	89 (59)	0.042 ***** 0.027 ^†^ 1.000 ^‡^	119 (78)	0.139 ***** 0.001 ^†^ 0.307 ^‡^	125 (28)	0.065 ***** 0.011 ^†^ 1.000 ^‡^	0.186; 0.086 ^T0^ <0.001; 0.461 ^T1^ 0.012; 0.275 ^T2^ 0.001; 0.405 ^T3^ 0.008; 0.293 ^T4^
	TPE	94 (37)		112 (31)		147 (68)		157 (30)		160.5 (23)		
	MPT	111 (41)		148 (45)		145 (53)		167 (21)		176 (39)		
4 m Walking Test (m/s)	Con	0.571 (0.321)	-	0.560 (0.336)	-	0.502 (0.498)	-	0.771 (0.173)	-	0.754 (0.255)	1.000 ***** 0.033 ^†^ 0.412 ^‡^	0.114; 0.111 ^T0^ 0.172; 0.095 ^T1^ 0.161; 0.118 ^T2^ 0.334; 0.073 ^T3^ 0.039; 0.197 ^T4^
	TPE	0.774 (0.381)		0.735 (0.352)		0.722 (0.506)		0.818 (0.517)		0.755 (0.415)		
	MPT	0.688 (0.283)		0.703 (0.323)		0.751 (0.386)		0.812 (0.398)		0.913 (0.435)		
YBT-A (cm)	Con	0 (24)	-	0 (10.9)	0.001 ***** 0.010 ^†^ 1.000 ^‡^	-		12.17 (28.3)	0.044 ***** 0.042 ^†^ 1.000 ^‡^	27 (30.9)	0.210 ***** 0.062 ^†^ 1.000 ^‡^	0.449; 0.041 ^T0^ <0.001; 0.421 ^T1^ -^T2^ 0.016; 0.278 ^T3^ 0.043; 0.185 ^T4^
	TPE	22.2 (32.8)		31.2 (13.7)				34 (19.8)		30.1 (22)		
	MPT	0 (26.8)		28.5 (9.9)				34 (10.8)		31.8 (20.3)		
YBT-M (cm)	Con	0 (0)	-	0 (20.7)	0.290 ***** 0.044 ^†^ 1.000 ^‡^	-		0 (47.5)	0.084 ***** 0.090 ^†^ 1.000 ^‡^	0 (48.4)	0.024 ***** 0.103 ^†^ 1.000 ^‡^	0.810; 0.011 ^T0^ 0.044; 0.169 ^T1^ -^T2^ 0.037; 0.221 ^T3^ 0.017; 0.246 ^T4^
	TPE	0 (24.6)		0 (56.6)				48.6 (31.1)		54.4(15.1)		
	MPT	0 (0)		42 (43.25)				53.7 (24.9)		50.5 (9)		
YBT-L (cm)	Con	0 (45)	-	0 (23.3)	0.131 ***** 0.025 ^†^ 1.000 ^‡^	-		19.8 (56.9)	-	0 (58.8)	0.097 ***** 0.015 ^†^ 1.000 ^‡^	0.907; 0.005 ^T0^ 0.022; 0.207 ^T1^ -^T2^ 0.154; 125 ^T3^ 0.012; 0.269 ^T4^
	TPE	0 (48.8)		44.3 (59.4)				54.8 (38.7)		58.1 (19.1)		
	MPT	0 (38.8)		52 (10.3)				54.7 (23)		64.7 (13.2)		

***** Control vs. TPE; ^† ^Control vs. MPT; ^‡^ TPE vs. MPT. Abbreviations: Con, Control Group; TPE, Therapeutic Patient Education Group; MPT, Multimodal Physiotherapy Group; IQR, Interquartile Range; PCS, Pain Catastrophizing Scale; VAS, Visual Analog Scale; WOMAC, Western Ontario and McMaster Universities Osteoarthritis Index; TSK-11, Tampa Scale of Kinesiophobia; CPSES, Chronic Pain Self-Efficacy Scale; YBT, Y-Balance Test (A, Anterior; M, Medial; L, Lateral).

### 3.2. Intragroup Comparison of Different Operative Periods (Table 3)

Each variable demonstrated a significant improvement from baseline (T0) to the six-month post-surgery follow-up (T4), except for CPSES in the control group. In the preoperative period, which assessed the immediate effects of the preoperative interventions in both experimental groups, the control group exhibited a statistically significant improvement in YBT-A. However, no other variables showed significant changes. Conversely, participants in the TPE group manifested significant improvements with large effect sizes in multiple variables, including PCS, VAS Rest, VAS Walk, WOMAC Pain, WOMAC total score, TSK-11, CPSES, YBT-A, and YBT-M. Similarly, participants in the MPT group displayed significant improvements and large effect sizes in a range of variables, including PCS, VAS Rest, VAS Walk, WOMAC Pain, WOMAC total score, TSK-11, CPSES, YBT-A, YBT-M, and YBT-L. During the perioperative period, which encompassed the interval from T1 to T2, the control group demonstrated significant improvements with large effect sizes in PCS, WOMAC Pain, and WOMAC total score. Participants in the TPE group exhibited significant improvements and large effect sizes in VAS Rest, VAS Walk, WOMAC Pain, and WOMAC total score. Additionally, participants in the MPT group showed significant improvements in VAS Walk, WOMAC Pain, and WOMAC total score. The postoperative period was bifurcated into two segments for the analysis. The first segment, assessing changes from T2 to T3, revealed significant improvements with large effect sizes in the control group for PCS, VAS Rest, WOMAC Pain, WOMAC total score, and 4 m Walking Test. Similar trends were observed in the TPE group for VAS Rest, WOMAC Pain, WOMAC total score, and 4 m Walking Test. The MPT group also showed significant improvements in PCS, VAS Rest, WOMAC Pain, WOMAC total score, CPSES, and 4 m Walking Test. In the second segment of the postoperative period, assessing changes from T3 to T4, no significant improvements were observed in the control group. However, the TPE group exhibited significant improvements in WOMAC Pain and WOMAC total score. The MPT group also displayed significant improvements in VAS Rest and YBT-L. A graphical representation of the results can be found in the Appendix A.

**Table 3 jcm-14-00268-t003:** Intragroup comparison results across preoperative, perioperative, and postoperative periods.

Variables	Groups	Preoperative Period	Perioperative Period	Postoperative Period		Friedmann’s Anova (*p*-Value)	Effect Size (r)
		Baseline vs. Post-Treatment ^*^	Post-Treatment vs. 1 Month Post-Surgery ^†^	1 Month Post-Surgery vs. 3 Months Post-Surgery ^‡^	3 Months Post-Surgery vs. 6 Months Post-Surgery ^§^		
PCS Total	Con	0.916	0.028	0.010	0.965	<0.001	0.029 *, 0.693 ^†^, 0.777 ^‡^, 0.013 ^§^
	TPE	0.002	0.068	0.066	0.236	<0.001	0.850 *, 0.550 ^†^, 0.554 ^‡^, 0.375 ^§^
	MPT	0.003	0.575	0.035	0.864	<0.001	0.848 *, 0.177 ^†^, 0.667 ^‡^, 0.052 ^§^
VAS Rest	Con	0.813	0.219	0.049	0.593	0.001	0.065 *, 0.370 ^†^, 0.545 ^‡^, 0.143 ^§^
	TPE	0.042	0.034	0.042	0.109	<0.001	0.564 *, 0.638 ^†^, 0.613 ^‡^, 0.507 ^§^
	MPT	0.004	0.779	0.046	0.027	<0.001	0.839 *, 0.089 ^†^, 0.632 ^‡^, 0.665 ^§^
VAS Walk	Con	0.752	0.075	0.171	0.430	<0.001	0.088 *, 0.537 ^†^, 0.379 ^‡^, 0.211 ^§^
	TPE	0.002	0.008	0.067	0.581	0.007	0.850 *, 0.804 ^†^, 0.551 ^‡^, 0.175 ^§^
	MPT	0.037	0.011	0.609	0.105	0.003	0.604 *, 0.807 ^†^, 0.162 ^‡^, 0.489 ^§^
WOMAC Pain	Con	0.937	0.033	0.015	0.630	<0.001	0.022 *, 0.643 ^†^, 0.673 ^‡^, 0.129 ^§^
	TPE	0.002	0.018	0.011	0.014	<0.001	0.839 *, 0.715 ^†^, 0.770 ^‡^, 0.778 ^§^
	MPT	0.009	0.026	0.011	0.440	<0.001	0.753 *, 0.703 ^†^, 0.802 ^‡^, 0.233 ^§^
WOMAC Total Score	Con	0.441	0.033	0.002	0.401	<0.001	0.214 *, 0.644 ^†^, 0.853 ^‡^, 0.224 ^§^
	TPE	0.021	0.008	0.003	0.013	<0.001	0.640 *, 0.805 ^†^, 0.888 ^‡^, 0.787 ^§^
	MPT	0.005	0.012	0.009	0.069	<0.001	0.805 *, 0.793 ^†^, 0.822 ^‡^, 0.549 ^§^
TSK-11	Con	0.624	0.153	0.533	0.754	0.046	0.136 *, 0.452 ^†^, 0.188 ^‡^, 0.091^§^
	TPE	0.004	0.798	0.284	10.000	0.005	0.801 *, 0.077^†^, 0.323^‡^, 0.000 ^§^
	MPT	0.004	0.260	0.609	0.502	0.001	0.840 *, 0.356 ^†^, 0.162 ^‡^, 0.203 ^§^
CPSES	Con	0.529	0.575	0.066	0.054	0.088	0.174 *, 0.177 ^†^, 0.554 ^‡^, 0.555^§^
	TPE	0.009	0.213	0.119	0.192	<0.001	0.727 *, 0.375 ^†^, 0.469 ^‡^, 0.413 ^§^
	MPT	0.007	0.445	0.007	0.539	<0.001	0.782 *, 0.242 ^†^, 0.855 ^‡^, 0.185 ^§^
4 m Walking Test	Con	0.650	0.114	0.007	0.424	0.019	0.126 *, 0.500 ^†^, 0.854 ^‡^, 0.241 ^§^
	TPE	0.861	0.203	0.008	0.508	0.009	0.049 *, 0.403 ^†^, 0.888 ^‡^, 0.210 ^§^
	MPT	0.530	0.445	0.012	0.214	0.018	0.181 *, 0.242 ^†^, 0.891 ^‡^, 0.415 ^§^
YBT-A	Con	0.043	-	-	0.116	0.008	0.561 *, 0.474 ^§^
	TPE	0.005			1.000	0.008	0.783 *, 0.000 ^§^
	MPT	0.022			0.869	0.005	0.662 *, 0.059 ^§^
YBT-M	Con	0.593	-	-	0.273	0.021	0.148 *, 0.330 ^§^
	TPE	0.046			0.066	<0.001	0.552 *, 0.581 ^§^
	MPT	0.008			0.401	<0.001	0.770 *, 0.280 ^§^
YBT-L	Con	0.223	-	-	0.138	0.004	0.338 *, 0.447 ^§^
	TPE	0.083			0.445	<0.001	0.481 *, 0.242 ^§^
	MPT	0.007			0.036	0.004	0.780 *, 0.700 ^§^

* Baseline vs. Post-treatment; ^†^ Post-treatment vs. 1 month post-surgery; ^‡^ 1 month post-surgery vs. 3 months post-surgery; ^§^ 3 months post-surgery vs. 6 months post-surgery. Abbreviations: Con, Control Group; TPE, Therapeutic Patient Education Group; MPT, Multimodal Physiotherapy Group; IQR, Interquartile Range; PCS, Pain Catastrophizing Scale; VAS, Visual Analog Scale; WOMAC, Western Ontario and McMaster Universities Osteoarthritis Index; TSK-11, Tampa Scale of Kinesiophobia; CPSES, Chronic Pain Self-Efficacy Scale; YBT, Y-Balance Test (A, Anterior; M, Medial; L, Lateral).

## 4. Discussion

The present three-armed, randomized controlled trial demonstrated that both preoperative interventions, therapeutic patient education (TPE) and multimodal physical therapy (MPT), effectively reduce pain catastrophizing in patients predisposed to this condition. Additionally, these interventions can improve pain during walking, health functioning, pain-related fear of movement, self-efficacy, and dynamic balance. MPT showed better immediate effects, while the TPE group had better long-term postoperative results compared to the control group. Different trends over time were observed between groups. Both experimental groups showed significant improvements in most variables during the preoperative and perioperative periods, with continued improvements in pain intensity and health functioning between the 3- and 6-month follow-up. The control group saw major improvements primarily in the first three months post-surgery.

In the last decade, addressing pain catastrophizing has been recommended by numerous authors to enhance postoperative outcomes after total knee arthroplasty (TKA) [8]. Interventions such as cognitive behavioral therapy, pain neuroscience education (PNE), hypnotic therapy, and therapeutic exercise have been investigated in this population, showing varying degrees of effectiveness in reducing pain catastrophizing [33]. These findings are consistent across different pain conditions, indicating that pain catastrophizing is a modifiable risk factor [34,35,36]. Our study supports these findings, as both experimental groups successfully reduced pain catastrophizing. However, the impact of reducing pain catastrophizing on postoperative outcomes following TKA remains unclear due to mixed results in the literature. For instance, Birch et al. [15] found no additive effect of cognitive behavioral therapy-based pain education on postoperative outcomes compared to usual care, and Riddle et al. [17] reported similar results for coping skills training. Conversely, our study observed that preoperative reduction in pain catastrophizing could improve postoperative outcomes.

PNE was essential in both experimental interventions in this study. It involves various educational methods aimed at changing patients’ understanding of pain, incorporating behavior change strategies, psychologically informed practices, and modern pain-related biological sciences to reduce fear, hypervigilance, anxiety, and worry about pain [37]. A study by Louw et al. [38] added a brief 30 min PNE session to usual care but found no postoperative benefits, possibly due to the short duration of the intervention and lack of screening for high pain catastrophizing. Our results show that a home-based intervention with three in-person PNE sessions led to better immediate outcomes.

Therapeutic exercise was also crucial in the preoperative interventions. In the TPE group, subjects performed exercises with minimal supervision, while the MPT group had a higher degree of supervision over two months. Despite conflicting evidence on the effectiveness of preoperative exercise for TKA patients, it generally improves postoperative function, and strength, and reduces the hospital stay, although not pain intensity [39,40]. These benefits are typically short-term, with inconclusive mid- and long-term effects [41].

In this study, combining therapeutic exercise with pain education maintained several benefits postoperatively, enhancing self-efficacy and reducing kinesiophobia, with high exercise adherence reported [19]. These findings are significant as self-efficacy mediates exercise adherence [42], and fear of movement can discourage physical activity in osteoarthritis patients [43]. Thus, patients receiving experimental interventions likely developed more adaptive behaviors perioperatively and postoperatively.

Exercise and patient education are first-line treatments for knee osteoarthritis, complemented by manual therapy for some patients [44]. Although with fewer than the recommended 12 sessions [44], the more supervised group had better immediate results in terms of pain catastrophizing, health functioning, self-efficacy, and dynamic balance. These results are consistent with early post-treatment benefits reported in other studies. Supervision and individualization are crucial, as they allow for adjustments based on patient progress, potentially enhancing treatment effectiveness and providing additional educational opportunities [44].

However, these effects often diminish over time, possibly due to low exercise adherence post-supervision [45]. Therefore, patient education programs should bolster self-efficacy and emphasize physical activity’s importance pre- and post-TKA. Our study showed that experimental groups had better self-efficacy, lower pain catastrophizing, and reduced pain-related fear of movement than the control group, likely contributing to sustained post-treatment results. We recommend assessing patient behaviors and cognitions throughout the perioperative process.

While the findings of this study demonstrate the efficacy of preoperative multimodal physiotherapy interventions in reducing pain catastrophizing and improving postoperative outcomes, their integration into routine healthcare systems presents several challenges. First, the logistics of delivering individualized, home-based sessions with varying levels of supervision could strain healthcare resources, particularly in systems with limited physiotherapy availability. Ensuring equitable access to such interventions would require careful workforce planning and potential upskilling of existing healthcare providers. Second, adherence to home-based interventions, while high in this study, might vary in real-world settings where patient engagement and self-motivation can be inconsistent. Third, the variability in session frequency and supervision levels highlights the need for standardization to optimize resource allocation while maintaining intervention efficacy.

Given these challenges, future studies should consider applying cost-effectiveness analyses to evaluate the financial feasibility of scaling these interventions. Such an analysis should compare the costs of implementing these programs with the potential savings from improved postoperative outcomes, such as reduced hospital stays, lower rates of chronic pain, and improved long-term functionality. Cost-effectiveness studies could also guide decisions on whether a higher degree of supervision, as seen in the MPT group, justifies the associated costs compared to minimally supervised programs like TPE. These considerations are crucial for informing policymakers and stakeholders about the potential for adopting such interventions in routine clinical practice, particularly in resource-constrained settings.

Finally, future research should also explore the implementation of other interventions that are critical for patients with osteoarthritis, such as weight management [46]. Weight control has been shown to play a significant role in the recovery of functional outcomes, including improvements in walking speed, which are essential for enhancing overall patient mobility and quality of life [47].

### Study Limitations

Several limitations should be noted. The sample size was relatively small and derived from a single hospital, which may limit the generalizability of the findings to broader populations. The trial was conducted during the COVID-19 pandemic, which impacted recruitment and surgery schedules. Some participants experienced longer preoperative periods due to surgery postponements and were followed up with monthly by phone to ensure adherence, with an additional session two weeks before surgery to reinforce key concepts. A second recruitment wave was conducted to meet the estimated sample size, resulting in slightly more participants than initially planned. Additionally, the use of self-reported questionnaires may have introduced recall bias or social desirability bias. Functional assessments, such as the Y-Balance Test, were not performed at all follow-up time points, limiting the interpretation of longitudinal comparisons. Despite these limitations, this study stands out for its innovative approach, targeting pain catastrophizing through preoperative home-based multimodal physiotherapy interventions. The randomized controlled trial design ensures methodological rigor, while the biopsychosocial framework integrates pain neuroscience education and therapeutic exercises, aligning with modern pain management models. Additionally, the six-month follow-up provides valuable insights into both short- and long-term effects, making the findings highly relevant for improving preoperative care in total knee arthroplasty.

## 5. Conclusions

The findings of this study show that preoperative physiotherapy effectively reduces pain catastrophizing in patients scheduled for TKA who catastrophize about their pain. Patients receiving preoperative physiotherapy showed better postoperative outcomes in pain catastrophizing, pain-related fear of movement, self-efficacy, and dynamic balance over a 6-month period. Despite better immediate results, more sessions and higher supervision did not lead to better long-term outcomes. These results support screening and targeting patients with high pain catastrophizing before TKA to enhance their postoperative pain-related outcomes, behaviors, and cognitions.

## Figures and Tables

**Figure 1 jcm-14-00268-f001:**
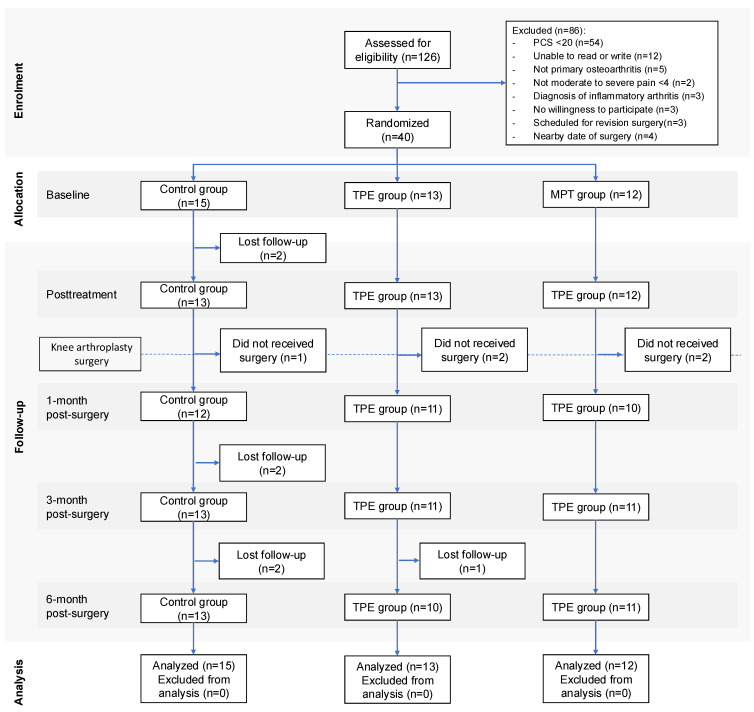
Flowchart of participants (CONSORT).

**Table 1 jcm-14-00268-t001:** Baseline sociodemographic and health characteristics of participants.

Population Description	Control (n = 15)	TPE (n = 13)	MPT (n = 12)	*p*-Value
Age (SD)	59.4 (6.16)	66.7 (5.18)	60.6 (5.80)	0.378 *****
Sex				0.537 ^†^
Male, n (%)	3 (20)	5 (38.5)	3 (25)	
Female, n (%)	12 (80)	8 (61.5)	9 (75)	
Body mass index				0.910 ^‡^
Normal [18.5–24.9], n (%)	1 (6.7)	-	-	
Overweight [25–29.9], n (%)	3 (20)	4 (30.8)	4 (33.3)	
Type I obesity [30–34.9], n (%)	7 (46.7)	9 (69.2)	6 (50)	
Type II obesity [35–39.9], n (%)	4 (26.7)	-	1 (8.3)	
Type III obesity [>40], n (%)	-	-	1 (8.3)	
Charlson comorbidity index				0.109 ^‡^
1, n (%)	-	-	-	
2, n (%)	3 (20)	3 (23.1)	5 (41.7)	
3, n (%)	4 (26.7)	7 (53.8)	7 (58.3)	
4, n (%)	4 (26.7)	2 (15.4)	-	
5, n (%)	4 (26.7)	1 (7.7)	-	
Smoking				0.269 ^‡^
Never smoked, n (%)	13 (86.7)	7 (53.8)	7 (58.3)	
Quit smoking, n (%)	1 (6.7)	6 (46.2)	5 (41.7)	
Smoker, n (%)	1 (6.7)	-	-	
Alcohol				0.757 ^‡^
Never, n (%)	7 (46.7)	6 (46.2)	4 (33.3)	
Minimal consumption, n (%)	6 (40)	5 (38.5)	7 (58.3)	
Usual consumption, n (%)	2 (13.3)	2 (15.4)	1 (8.3)	
Education level				0.133 ^‡^
Read and write, n (%)	7 (46.7)	4 (30.8)	1 (8.3)	
Elementary intermediate, n (%)	7 (46.7)	5 (38.5)	10 (83.3)	
Secondary vocational, n (%)	1 (6.7)	3 (23.1)	1 (8.3)	
University, n (%)	-	1 (7.7)	-	

***** Kruskal–Wallis Test; ^†^ Pearson’s Chi-squared Test; ^‡ ^Linear Chi-squared Test. Abbreviations: TPE, Therapeutic Patient Education Group; MPT, Multimodal Physiotherapy Group.

## Data Availability

All data are available upon reasonable request.

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
