# Peer review of "Preoperative Home-Based Multimodal Physiotherapy in Patients Scheduled for a Knee Arthroplasty Who Catastrophize About Their Pain: A Randomized Controlled Trial"

_jcm, 2025, doi:10.3390/jcm14010268_

Round 1

Reviewer 1 Report

Comments and Suggestions for Authors

This RCT study comparing preoperative multimodal physical therapy interventions for high-catastrophizing TKA patients. They evaluated different intervention intensities with 6-month follow-up. Despite sound methodology and high clinical significance, presentation quality needs improvement. Research demonstrates strong scientific merit with practical clinical applications. I have some comments:

1. Introduction section requires substantial strengthening by incorporating more recent systematic reviews and meta-analyses. While the current background provides basic information about chronic post-operative pain and pain catastrophizing, it would benefit significantly from updated references published within the last 3-5 years. This would better contextualize your research within the current scientific landscape and strengthen the theoretical foundation for your intervention choices.

2. Methodology section needs more detailed elaboration regarding the assessor training protocol and standardization procedures. While the randomization and intervention protocols are well-described, the lack of specific information about how assessors were trained and how intervention fidelity was maintained represents a significant gap. Including a clear description of the quality control measures and standardization processes would enhance the study's reproducibility and methodological rigor.

3. The results presentation, particularly Table 2, which is currently overly complex and difficult to interpret. Consider breaking down this table into smaller, more focused tables or supplementing it with visual representations such as graphs or charts. This would make the findings more accessible and help readers better understand the intervention effects across different time points.

4. Statistical analysis section would benefit from a more thorough explanation of how missing data were handled and the rationale for the chosen analytical methods. While the basic statistical approaches are appropriate, adding sensitivity analyses and more detailed justification for the selected statistical methods would strengthen the paper's scientific soundness and address potential reviewer concerns about the robustness of your findings.

5. Discussion section needs to better address the study limitations and provide more concrete implications for clinical practice. While the current discussion covers basic points, it should more critically evaluate the potential implementation challenges in real-world settings and provide specific recommendations for clinicians who might want to adopt this intervention approach.

6. Additionally, including a more detailed cost-effectiveness analysis would enhance the practical value of your findings for healthcare decision-makers.

Author Response

Reviewer 1:

This RCT study comparing preoperative multimodal physical therapy interventions for high-catastrophizing TKA patients. They evaluated different intervention intensities with 6-month follow-up. Despite sound methodology and high clinical significance, presentation quality needs improvement. Research demonstrates strong scientific merit with practical clinical applications. I have some comments:

  1. Introduction section requires substantial strengthening by incorporating more recent systematic reviews and meta-analyses. While the current background provides basic information about chronic post-operative pain and pain catastrophizing, it would benefit significantly from updated references published within the last 3-5 years. This would better contextualize your research within the current scientific landscape and strengthen the theoretical foundation for your intervention choices.

Authors: Thank you very much for your valuable suggestions. In response, we have incorporated the following recent systematic reviews and meta-analyses to enhance the discussion on the association between pain catastrophizing, postoperative function, and chronic post-surgical pain. These additions aim to better contextualize our research within the current scientific landscape:

Ashoorion V, Sadeghirad B, Wang L, et al. Predictors of Persistent Post-Surgical Pain Following Total Knee Arthroplasty: A Systematic Review and Meta-Analysis of Observational Studies. Pain Med. 2023;24(4):369-381. doi:10.1093/pm/pnac154

Sorel JC, Veltman ES, Honig A, Poolman RW. The influence of preoperative psychological distress on pain and function after total knee arthroplasty: a systematic review and meta-analysis. Bone Joint J. 2019;101-B(1):7-14. doi:10.1302/0301-620X.101B1.BJJ-2018-0672.R1

We appreciate your insightful feedback and are confident that these updates significantly strengthen the theoretical foundation of our study.

  1. Methodology section needs more detailed elaboration regarding the assessor training protocol and standardization procedures. While the randomization and intervention protocols are well-described, the lack of specific information about how assessors were trained and how intervention fidelity was maintained represents a significant gap. Including a clear description of the quality control measures and standardization processes would enhance the study's reproducibility and methodological rigor.

Authors: In response to the reviewer’s valuable suggestion, the following information has been incorporated into the methodology section:

“The assessor underwent training prior to the study to ensure strict adherence to standardized protocols. Although most variables were collected using self-reported questionnaires, all items were thoroughly reviewed, and standardized procedures were implemented to clarify participants' questions and provide feedback consistently. The assessments were conducted by a single physiotherapist with extensive experience in these procedures, ensuring uniformity and eliminating variability between assessors. Additionally, periodic reviews of assessment procedures were performed to maintain quality control and ensure compliance with the established protocols.”

  1. The results presentation, particularly Table 2, which is currently overly complex and difficult to interpret. Consider breaking down this table into smaller, more focused tables or supplementing it with visual representations such as graphs or charts. This would make the findings more accessible and help readers better understand the intervention effects across different time points.

Authors: We sincerely thank the reviewer for their thoughtful suggestion regarding the presentation of results in Table 2. We understand the potential challenge in interpreting such a comprehensive table and have carefully considered alternative approaches. While we explored breaking it down into smaller tables, we found that this resulted in an even more complex presentation and made it more difficult to cross-reference the data with ease.

After extensive discussions within the research team and aiming to address the reviewer’s concern, we have decided to retain the current format of Table 2. However, in response to the suggestion, we have supplemented the results with graphical representations for each variable, which we believe significantly enhance the clarity and accessibility of the findings. These graphs have been added as Supplementary Material to avoid overcrowding the main manuscript while still providing readers with a clear visual summary of the intervention effects over time.

It is worth noting that the Y Balance Test was not assessed at the one-month postoperative time point due to implant security considerations, as advised by surgeons. Consequently, a graphical representation for this variable was not included, as it would not have been as informative as for the other variables.

We believe that these additions improve the overall readability and comprehension of the results. To ensure that readers are aware of these supplementary materials, the following sentences have been added to the results section in both the between-group and within-group comparisons:

“A graphical representation of the results can be found in the Supplementary Materials (Supplementary Figures S1-S8).”

Additionally, the following information has been included at the end of the manuscript:

“Supplementary Materials: The following supporting information can be downloaded at: XXXX, Supplementary Figures S1-S8.”

We hope that these enhancements align with the reviewer’s recommendation and contribute to improving the overall quality and clarity of the manuscript. We are grateful for the valuable feedback that has allowed us to refine our presentation.

  1. Statistical analysis section would benefit from a more thorough explanation of how missing data were handled and the rationale for the chosen analytical methods. While the basic statistical approaches are appropriate, adding sensitivity analyses and more detailed justification for the selected statistical methods would strengthen the paper's scientific soundness and address potential reviewer concerns about the robustness of your findings.

Authors: We thank the reviewer for highlighting the importance of providing additional justification for our chosen statistical methods. The decision to use nonparametric tests was based on the results of the Shapiro-Wilk test, which indicated that the data did not follow a normal distribution. Nonparametric methods, such as the Kruskal-Wallis test and Friedman's ANOVA, were selected as they are robust, appropriate for small sample sizes, and well-suited for skewed data distributions.

To enhance the interpretability of our findings, we included effect size calculations (epsilon-squared and r) to quantify the magnitude of differences observed, providing a clearer understanding of their clinical relevance. Additionally, missing data were minimal and handled by analyzing complete cases to ensure the integrity of the results.

Given the robustness of nonparametric methods and the straightforward nature of our dataset, sensitivity analyses were deemed unnecessary. We believe these methods ensure the reliability and validity of our findings while aligning with the study’s design and objectives.

  1. Discussion section needs to better address the study limitations and provide more concrete implications for clinical practice. While the current discussion covers basic points, it should more critically evaluate the potential implementation challenges in real-world settings and provide specific recommendations for clinicians who might want to adopt this intervention approach.

Authors: We sincerely thank the reviewer for their valuable feedback and the opportunity to improve our manuscript. In response, we have reworked the study limitations section to provide a more comprehensive evaluation:

“Several limitations should be noted. The sample size was relatively small and derived from a single hospital, which may limit the generalizability of the findings to broader populations. The trial was conducted during the COVID-19 pandemic, which impacted recruitment and surgery schedules. Some participants experienced longer preoperative periods due to surgery postponements and were followed up monthly by phone to ensure adherence, with an additional session two weeks before surgery to reinforce key concepts. A second recruitment wave was conducted to meet the estimated sample size, resulting in slightly more participants than initially planned. Additionally, the use of self-reported questionnaires may have introduced recall bias or social desirability bias. Functional assessments, such as the Y Balance Test, were not performed at all follow-up time points, limiting the interpretation of longitudinal comparisons.”

Additionally, we have expanded the discussion section to critically evaluate the implementation challenges of the intervention in real-world settings and to provide specific recommendations for clinical practice:

“While the findings of this study demonstrate the efficacy of preoperative multimodal physiotherapy interventions in reducing pain catastrophizing and improving postoperative outcomes, their integration into routine healthcare systems presents several challenges. First, the logistics of delivering individualized, home-based sessions with varying levels of supervision could strain healthcare resources, particularly in systems with limited physiotherapy availability. Ensuring equitable access to such interventions would require careful workforce planning and potential upskilling of existing healthcare providers. Second, adherence to home-based interventions, while high in this study, might vary in real-world settings where patient engagement and self-motivation can be inconsistent. Third, the variability in session frequency and supervision levels highlights the need for standardization to optimize resource allocation while maintaining intervention efficacy.”

We hope these revisions address the reviewer’s concerns and enhance the quality of the manuscript by providing a more critical evaluation of the study limitations and concrete implications for clinical practice. We greatly appreciate the reviewer’s insights, which have helped us strengthen our discussion and improve the manuscript’s relevance for both researchers and clinicians.

  1. Additionally, including a more detailed cost-effectiveness analysis would enhance the practical value of your findings for healthcare decision-makers.

Authors: We sincerely agree with the reviewer on the importance of conducting a cost-effectiveness analysis to enhance the practical value of these findings for healthcare decision-makers. While it was not feasible to perform such an analysis in the present study, we have incorporated a section at the end of the discussion to emphasize the importance of cost-effectiveness studies in future research. The added text reads as follows:

“Given these challenges, future studies should consider applying cost-effectiveness analyses to evaluate the financial feasibility of scaling these interventions. Such an analysis should compare the costs of implementing these programs with the potential savings from improved postoperative outcomes, such as reduced hospital stays, lower rates of chronic pain, and improved long-term functionality. Cost-effectiveness studies could also guide decisions on whether a higher degree of supervision, as seen in the MPT group, justifies the associated costs compared to minimally supervised programs like TPE. These considerations are crucial for informing policymakers and stakeholders about the potential for adopting such interventions in routine clinical practice, particularly in resource-constrained settings.”

We hope this addition aligns with the reviewer’s suggestion and reinforces the practical implications of the study for future applications

Reviewer 2 Report

Comments and Suggestions for Authors

Discussion: Thanks to the authors for addressing such an interesting and timely topic.

Abstract: Well-organized, it provides an overview of the article, and the aim of the study is clear.

Introduction: Overall, this section is well-written. However, the authors should add more on about pain catastrophizing, providing a more detailed explanation of its definition and exploring the insights offered by the existing literature on this topic.

Methods: The methodological approach is sound, but the section on "Interventions" lacks clarity. The authors should offer a more detailed description of the three different groups, specifying their characteristics and the specific protocol provided to each group.

Results: The results are clear and well-structured. The tables are particularly useful and effectively highlight the key results. One point that remains unclear, however, is the post-operative protocol. Did the surgeons follow the same post-operative guidelines (e.g., weight-bearing allowed on the first post-op day, or any other specific instructions)? 

Results: what are the strenghts of this article and what suggestion authors want to give to sugerons in clinical practice?

Author Response

Reviewer 2:

Discussion: Thanks to the authors for addressing such an interesting and timely topic.

Abstract: Well-organized, it provides an overview of the article, and the aim of the study is clear.

Introduction: Overall, this section is well-written. However, the authors should add more on about pain catastrophizing, providing a more detailed explanation of its definition and exploring the insights offered by the existing literature on this topic.

Authors: We sincerely thank the reviewer for their thoughtful suggestion, which has allowed us to further improve our introduction by incorporating additional insights into the construct of pain catastrophizing. This feedback has provided us with an opportunity to delve deeper into the existing literature and explore perspectives we were not previously aware of. As a result, the following text has been added to the introduction:

“Pain catastrophizing has been conceptualized as the tendency to magnify the threat value of a pain stimulus, to feel helpless in the context of pain, and as a relative inability to inhibit pain-related thoughts in anticipation of, during, or following a painful event [10]. However, a recent study by Petrini and Arendt-Nielsen proposed a re-conceptualization of this construct within a modern biopsychosocial framework. They described it as an emotional regulation strategy and a form of catastrophic worry (characterized by repetitive negative thinking), arising from the interplay between a ruminative process and personality trait characteristics associated with the behavioral inhibition system [11].”

We are grateful for the reviewer’s comments, which have significantly enhanced the depth and clarity of our introduction. We hope this addition addresses the suggestion and enriches the manuscript’s conceptual framework.

Methods: The methodological approach is sound, but the section on "Interventions" lacks clarity. The authors should offer a more detailed description of the three different groups, specifying their characteristics and the specific protocol provided to each group.

Authors: We appreciate the reviewer’s observation regarding the need for more detail in the "Interventions" section. We understand how it might appear that this section lacks information; however, the interventions have been extensively described in the feasibility study referenced in the first line of this section:

Terradas-Monllor M, Ochandorena-Acha M, Beltran-Alacreu H, Garcia Oltra E, Collado Saenz F, Hernandez Hermoso J. A feasibility study of home-based preoperative multimodal physiotherapy for patients scheduled for a total knee arthroplasty who catastrophize about their pain. Physiother Theory Pract. 2023;39(8):1606-1625. doi:10.1080/09593985.2022.2044423.

Despite this, we aimed to present the essential information to provide an overview of the interventions in the three groups while maintaining the primary focus on the results, which were complex to write and interpret. This approach was chosen to enhance the overall readability of the manuscript and ensure that the study's core findings were clearly communicated.

We hope the reviewer understands the rationale behind this decision by the research team. However, we remain open to providing further clarification or additional details about the interventions if the current description is deemed insufficient.

Results: The results are clear and well-structured. The tables are particularly useful and effectively highlight the key results. One point that remains unclear, however, is the post-operative protocol. Did the surgeons follow the same post-operative guidelines (e.g., weight-bearing allowed on the first post-op day, or any other specific instructions)? 

Authors: We sincerely thank the reviewer for their thoughtful comment, which has helped us identify the need for greater clarity in describing the post-operative protocol. In the "Interventions" section, under the control group description, we detail that participants received early inpatient physiotherapy and home-based postoperative physiotherapy. Early physiotherapy focused on early mobilization and ambulation, beginning the same day as the surgical intervention and continuing until discharge, typically around four days post-surgery. After discharge, a physiotherapist was assigned to each patient through an external domiciliary rehabilitation service, which generally started within one to ten days post-discharge.

We acknowledge that this information, while detailed in our previous feasibility study, could benefit from further elaboration in this manuscript. As a result, we have revised this section to provide additional clarity in response to the reviewer’s concerns. The updated text now reads:

“Participants allocated to the control group received usual care, which included a multidisciplinary, group-based, biomedical preoperative education session and early inpatient physiotherapy focused on early mobilization and ambulation. This phase began on the day of the surgical intervention and continued until discharge, typically around four days post-surgery. Following discharge, each patient was assigned a physiotherapist through an external domiciliary rehabilitation service, which generally started within one to ten days after discharge. This was followed by home-based postoperative physiotherapy.”

We hope that this revision addresses the reviewer’s concern and enhances the clarity of the manuscript. We are truly grateful for the reviewer’s insightful feedback, which has allowed us to improve the quality of the paper.

Results: what are the strenghts of this article and what suggestion authors want to give to sugerons in clinical practice?

Authors: We sincerely thank the reviewer for their insightful comments, which have allowed us to further enhance the manuscript. In response, we have added the following information immediately after the limitations section, highlighting the strengths of the study:

“Despite these limitations, this study stands out for its innovative approach, targeting pain catastrophizing through preoperative home-based multimodal physiotherapy interventions. The randomized controlled trial design ensures methodological rigor, while the biopsychosocial framework integrates pain neuroscience education and therapeutic exercises, aligning with modern pain management models. Additionally, the six-month follow-up provides valuable insights into both short- and long-term effects, making the findings highly relevant for improving preoperative care in total knee arthroplasty.”

Furthermore, we would like to clarify that the conclusions of the study include the following statement:

“These results support screening and targeting patients with high pain catastrophizing before TKA to enhance their postoperative pain-related outcomes, behaviors, and cognitions.”

This recommendation serves as a suggestion for both physiotherapists and orthopedic surgeons, emphasizing the importance of identifying and addressing pain catastrophizing in patients undergoing total knee arthroplasty to optimize postoperative recovery.

Reviewer 3 Report

Comments and Suggestions for Authors

Introduction:

Pag 2 (line 54-55): “Few studies have tested this hypothesis; Birch S. et al. 54 (2020) and Buvanendran A. et al. (2021)….” Please review the citation style.

Throughout the study, the authors use the term ‘therapeutic exercise’ and ‘exercise’ as synonyms. Are they the same? We recommend that they be homogeneous in the description.

2.1 Design

Why do the authors provide two records from two different ethics committees?

Figure 1: It should be restructured to facilitate the reader's understanding. The font size is small

Weight loss is known to be one of the gold standards for the management of osteoarthritis of the knee. Did participants have weight control during the study?

It indicates as a criterion the achievement of a score of 20 points or more in PCS, and uses an appointment in patients with fibromyalgia. Justify this

Pag 2 (line 88-89): “(vi) were able to read and understand Spanish or Catalan”.

This can be a problem. If participants cannot read or understand Spanish, there may be losses in the study that have not been taken into account, as the measurement instruments used are validated in Spanish.

2.5 Sample size calculation. We recommend that the authors include paragraph 2.5 after paragraph 2.2.

A procedures section is missing, as it is not known how participants reach the study investigators. It is also not known who makes the diagnosis of osteoarthritis, what degree of severity they present etc...

Do all patients live in the same locality? Do they come from the same hospital unit?

Interventions section:

It is recommended that you describe the intervention in a structured way for each group. What was the support used for the therapeutic home-based education programme? What was the duration of each session? How are the sessions structured? Delivering this type of intervention for patients with the level of education described in table 1 is difficult. The authors should explain this in this study.

Similarly with the group receiving the multimodal physiotherapy programme, how and by whom is the orthopaedic manual therapy performed? How long? Regarding the exercise applied, indicate periodicity, frequency, series, volume etc. according to Tidier guidelines.

2.4. Outcome measures

Dynamic balance is assessed with the Y Balance Test.

What was the criterion for its choice instead of others more commonly used in the study population?

Results:

With regard to educational level, there are also notable differences within each group. This may be relevant to the results achieved. Discuss this aspect.

Table 2 (pag 7: line 215): Given that catastrophic pain is a primary variable, why do the authors not report results on the dimensions measured by this instrument? With the WOMAC questionnaire they do.

Discusion:

Pag 11: line 305-309: “Although fewer than the recommended 12 sessions [43], the more supervised group had better immediate results, consistent with early post-treatment benefits seen in other studies” 

Authors need to be more specific. What specific results are they referring to? What are the physiological mechanisms that may explain the observed improvements?

Pag 11: line 294-295: “These benefits are typically short-term, with inconclusive mid and long-term effects [40]”.

In the present study, the therapeutic exercise is not applied in isolation in any of the groups. The authors should be thorough in the explanations provided, as this is a relevant aspect of the study.

Author Response

Reviewer 3:

Introduction:

Pag 2 (line 54-55): “Few studies have tested this hypothesis; Birch S. et al. 54 (2020) and Buvanendran A. et al. (2021)….” Please review the citation style.

Autors: We sincerely thank the reviewer for bringing this to our attention. The references mentioned on page 2 (lines 54–55) and additional citations in the discussion section have been carefully reviewed and adjusted to ensure they adhere to the correct citation style. We greatly appreciate your meticulous attention to detail, which has allowed us to further refine the manuscript.

Throughout the study, the authors use the term ‘therapeutic exercise’ and ‘exercise’ as synonyms. Are they the same? We recommend that they be homogeneous in the description.

Authors: Yes, we wanted to mean the same (therapeutic exercise), we have adapted the text and made it more homogeneous. Thank you for your suggestion.

2.1 Design

Why do the authors provide two records from two different ethics committees?

Authors: We thank the reviewer for this question, which provides an opportunity to clarify the ethical review process for this study. Initially, the study was assessed by the Research Ethics Committee of the University of Vic – Central University of Catalonia to meet the requirements for a funding opportunity, as it was necessary to have an ethics committee evaluate and approve the project at that stage. Subsequently, to initiate the project at the Hospital Germans Trias i Pujol, it was also required that the Human Research Ethics Committee of the hospital review and approve the study. Since the project had already been approved by the first committee, the subsequent approval process at the hospital was expedited.

Figure 1: It should be restructured to facilitate the reader's understanding. The font size is small.

Authors: We appreciate your feedback regarding Figure 1. To address your concerns, we have restructured the figure to enhance its clarity and make it more intuitive for readers. Additionally, we have increased the font size to improve readability. The updated figure has been included in the revised manuscript as Figure 1.

Thank you for bringing this to our attention, and we hope the revised version meets your expectations.

Weight loss is known to be one of the gold standards for the management of osteoarthritis of the knee. Did participants have weight control during the study?

Authors: We sincerely thank the reviewer for raising this important point. We fully agree that weight control could have been a valuable addition to our preoperative intervention. Unfortunately, it was not possible to include this component due to the lack of a nutritionist on our research team at the time. However, we recognize the significant role that weight management plays in the management of knee osteoarthritis and have highlighted this limitation in the discussion section.

To address this, we have added the following paragraph at the end of the discussion:

“Finally, future research should also explore the implementation of other interventions that are critical for patients with osteoarthritis, such as weight management [46]. Weight control has been shown to play a significant role in the recovery of functional outcomes, including improvements in walking speed, which are essential for enhancing overall patient mobility and quality of life [47].”

We hope this addition acknowledges the importance of weight control and provides a clear direction for future research. We greatly appreciate the reviewer’s insight, which has allowed us to further strengthen the manuscript.

It indicates as a criterion the achievement of a score of 20 points or more in PCS, and uses an appointment in patients with fibromyalgia. Justify this.

Authors: We thank the reviewer for raising this point, and we would like to provide further clarification. The reference cited immediately after the criterion refers to the Spanish validation of the Pain Catastrophizing Scale (PCS). The criterion of 20 points on the PCS was determined based on a series of preliminary studies conducted with our population. This threshold was established to identify individuals with moderate to severe levels of pain catastrophizing, as advised by Sullivan et al. in the PCS User’s Manual.

Additionally, the full description of the criteria and methodology used to set this cutoff is available in our previously published feasibility study, as mentioned at the beginning of the “Participants” section. For reference, this is the relevant information from the feasibility study:

“The cutoff for PCS was 20 (range 0–52) according to the 50th percentile on data collected from 40 consecutive participants before the study started. The decision for the cutoff setting was based on the PCS – User Manual by Sullivan, Bishop, and Pivik (2009). Individuals who score between the 50th and 75th percentiles on the PCS are considered at moderate risk for the development of chronicity, and those who score above the 75th percentile are considered at high risk (Sullivan, Bishop, and Pivik, 2009).”

We hope this explanation clarifies the rationale behind the chosen cutoff and its alignment with validated guidelines and previous research. We appreciate the reviewer’s attention to this detail, as it allowed us to provide further justification for our methodology.

Pag 2 (line 88-89): “(vi) were able to read and understand Spanish or Catalan”. This can be a problem. If participants cannot read or understand Spanish, there may be losses in the study that have not been taken into account, as the measurement instruments used are validated in Spanish.

Authors: Thank you for raising this concern. The criterion regarding the ability to read and understand Spanish or Catalan was included to account for potential participants who might not speak either language. However, we would like to clarify that, in this study, all participants were fluent in Spanish, ensuring that there were no losses related to the language of the measurement instruments.

This criterion was established as a safeguard but did not pose any issues in the present study since all participants could fully understand and complete the instruments validated in Spanish. We hope this clarification addresses your concern.

2.5 Sample size calculation. We recommend that the authors include paragraph 2.5 after paragraph 2.2.

Authors: We thank the reviewer for their suggestion. As recommended, the sample size calculation section has been moved and is now placed immediately after the "Participants" section to improve the logical flow of the manuscript.

A procedures section is missing, as it is not known how participants reach the study investigators. It is also not known who makes the diagnosis of osteoarthritis, what degree of severity they present, etc...

Authors: We sincerely thank the reviewer for pointing out this missing information. It is true that details regarding who performed the diagnosis and reviewed the inclusion and exclusion criteria were not included in the initial manuscript. We have now addressed this oversight by adding the following information to the manuscript:

“Orthopedic surgeons (EGA, FAO, and JHH) assessed participants for inclusion once they were scheduled for TKA.”

We appreciate the reviewer’s attention to this detail, which has allowed us to improve the clarity and completeness of the methods section.

Do all patients live in the same locality? Do they come from the same hospital unit?

Authors: Yes, all participants in the study lived in the same locality and were recruited from the same hospital unit. As stated in the manuscript (Section 2.2, Participants), the information regarding participants’ provenance is detailed in a previously conducted feasibility study, which we referenced to provide additional context.

We appreciate your attention to this detail, and we hope this clarification is helpful.

Interventions section:

It is recommended that you describe the intervention in a structured way for each group. What was the support used for the therapeutic home-based education programme? What was the duration of each session? How are the sessions structured? Delivering this type of intervention for patients with the level of education described in table 1 is difficult. The authors should explain this in this study. Similarly with the group receiving the multimodal physiotherapy programme, how and by whom is the orthopaedic manual therapy performed? How long? Regarding the exercise applied, indicate periodicity, frequency, series, volume etc. according to Tidier guidelines.

Authors: We sincerely thank the reviewer for their thoughtful comments and the opportunity to clarify this aspect of our study. The structured explanation, including the support used for the therapeutic home-based education program, the duration of each session, the session structure, and the information about whom the orthopedic manual therapy was performed has been extensively detailed in a previously published feasibility study:

Terradas-Monllor M, Ochandorena-Acha M, Beltran-Alacreu H, Garcia Oltra E, Collado Saenz F, Hernandez Hermoso J. A feasibility study of home-based preoperative multimodal physiotherapy for patients scheduled for a total knee arthroplasty who catastrophize about their pain. Physiother Theory Pract. 2023;39(8):1606-1625. doi:10.1080/09593985.2022.2044423.

In this manuscript, we aimed to present the essential information to provide an overview of the interventions for the three groups while maintaining the primary focus on the results, which were complex to write and interpret. This approach was chosen to enhance the overall readability of the manuscript and ensure the study’s core findings were clearly communicated.

We hope the reviewer understands the rationale behind this decision by the research team. However, we remain open to providing further clarification or additional details about the interventions if the current description is considered insufficient.

2.4. Outcome measures

Dynamic balance is assessed with the Y Balance Test.

What was the criterion for its choice instead of others more commonly used in the study population?

Authors: We thank the reviewer for this insightful question. The Y Balance Test (YBT) was chosen for several reasons. First, it is a well-validated tool for assessing dynamic balance, particularly in populations with lower-limb impairments, such as patients undergoing TKA. Second, the YBT provides a standardized and quantifiable measure of balance that is sensitive to functional limitations and improvements over time. Compared to other balance tests, the YBT evaluates the ability to maintain balance while performing controlled reach movements, which closely mimic functional tasks required in daily activities, such as walking and reaching.

Additionally, the YBT requires minimal equipment, is relatively easy to administer, and can be performed safely in both clinical and home-based settings, making it a practical choice for this study population. Although other tests are also commonly used, the YBT’s focus on dynamic balance aligns with the study's objective of assessing functional recovery and mobility-related outcomes in patients with osteoarthritis undergoing TKA.

We hope this explanation clarifies the rationale behind the choice of the YBT for this study.

Results:

With regard to educational level, there are also notable differences within each group. This may be relevant to the results achieved. Discuss this aspect.

Authors: We thank the reviewer for this important observation. According to our results, no significant differences in educational level were observed between the groups. However, we acknowledge the potential relevance of educational level to the outcomes achieved, particularly in how therapists adapt their explanations of pain to individual participants. This aspect has been explored in a previous qualitative study, which discussed the influence of communication styles and adaptations made by therapists to accommodate participants’ limitations.

Ochandorena-Acha M, Escribà-Salvans A, Hernández-Hermoso JA, Terradas-Monllor M. Perioperative experiences of patients with high pain catastrophizing and knee arthroplasty after receiving or not preoperative physiotherapy: Qualitative study. Musculoskelet Sci Pract. 2024 Apr;70:102918. doi: 10.1016/j.msksp.2024.102918. Epub 2024 Feb 2. PMID: 38330866.

In this study, the physiotherapist who performed all interventions did not report educational level as a limiting factor in delivering the intervention. Therefore, the research team considered that it was not necessary to highlight this aspect further in the current manuscript.

Table 2 (pag 7: line 215): Given that catastrophic pain is a primary variable, why do the authors not report results on the dimensions measured by this instrument? With the WOMAC questionnaire they do.

Authors: We thank the reviewer for this thoughtful comment. The WOMAC Pain subscale was included in the tables because the Visual Analog Scale (VAS) can sometimes be challenging to assess in older populations, and the WOMAC Pain subscale provides valuable information about pain specifically related to lower extremity function.

While we recognize the importance of reporting the PCS subscales, we aimed to maintain clarity and readability in the tables, given the large number of variables already included. Furthermore, our primary objective was to evaluate and improve the total PCS score, as this was the main variable of interest in our study.

We hope this explanation clarifies our rationale, but we remain open to suggestions on how to better present the data if necessary.

Discussion:

Pag 11: line 305-309: “Although fewer than the recommended 12 sessions [43], the more supervised group had better immediate results, consistent with early post-treatment benefits seen in other studies”. Authors need to be more specific. What specific results are they referring to? What are the physiological mechanisms that may explain the observed improvements?

Authors: We sincerely thank the reviewer for this valuable comment, which allowed us to clarify and expand on the specific results and their underlying mechanisms. Following the reviewer’s suggestion, we have added the specific outcomes we were referring to and provided an evidence-based rationale for the observed differences:

“Although fewer than the recommended 12 sessions [44], the more supervised group demonstrated better immediate results in terms of pain catastrophizing, health functioning, self-efficacy, and dynamic balance. These results are consistent with early post-treatment benefits reported in other studies. Supervision and individualization are crucial, as they allow for adjustments based on patient progress, potentially enhancing treatment effectiveness and providing additional educational opportunities [44].”

We hope this addition addresses the reviewer’s concern and provides greater clarity regarding the findings and their interpretation.

Pag 11: line 294-295: “These benefits are typically short-term, with inconclusive mid and long-term effects [40]”. In the present study, the therapeutic exercise is not applied in isolation in any of the groups. The authors should be thorough in the explanations provided, as this is a relevant aspect of the study.

Authors: We thank the reviewer for this insightful comment, which highlights an important aspect of our study. In the present study, therapeutic exercise was indeed not applied in isolation but rather as part of a multimodal intervention strategy. In both experimental groups, exercise was combined with therapeutic patient education and other components aimed at addressing pain catastrophizing and improving functional outcomes. This integrative approach aligns with the biopsychosocial model of pain management and reflects current best practices for managing knee osteoarthritis.

The statement in question refers to the existing evidence on therapeutic exercise applied in isolation, which typically shows short-term benefits but inconclusive mid- and long-term effects. In our study, the combination of therapeutic exercise with additional interventions likely contributed to the observed improvements in outcomes such as pain catastrophizing, self-efficacy, and dynamic balance. This highlights the importance of a comprehensive, multimodal approach, rather than relying solely on exercise. We have clarified this aspect in the manuscript to ensure it is accurately represented.

Round 2

Reviewer 1 Report

Comments and Suggestions for Authors

The authors have addressed my comments well and revised their manuscript adequately. Thank you for your effort.

Author Response

The authors have addressed my comments well and revised their manuscript adequately. Thank you for your effort.

Authors: We sincerely thank the reviewer for their kind feedback and for recognizing the effort we have put into revising the manuscript. We deeply appreciate your thoughtful comments, which have greatly contributed to improving the quality and clarity of our work.

Reviewer 3 Report

Comments and Suggestions for Authors

I appreciate the reviews made by the authors, however, I consider that there are two aspects that still need to be improved:

The explanation regarding language proficiency remains unclear. I recommend modifying item vi (section 2.2) as follows: (vi) were able to read and understand the Spanish language.

On the other hand, the reproducibility of the intervention must be guaranteed in a scientific study. This reviewer considers it relevant that this study has sufficient information on the type of intervention performed. It is not enough with this sentence: 

line 134-136: ‘Therapeutic education was based on PNE and CST, whilst therapeutic exercise was composed of strengthening, balance and stability exercises, and divided into core, hip, knee, and global exercises’.

Readers should not have to consult another previous study by the authors to understand what has been done in the present study.

Author Response

Reviewer 3:

Comment 1: I appreciate the reviews made by the authors, however, I consider that there are two aspects that still need to be improved:

Authors: We sincerely thank the reviewer for their observation regarding the clarity of the language proficiency criterion. Following the reviewer’s suggestion, we have revised item (vi) in section 2.2 to ensure greater precision. The updated text now reads:

“…were able to read and understand the Spanish language.”

We appreciate the reviewer’s input, which has helped us improve the clarity and accuracy of this section.

Comment 2: On the other hand, the reproducibility of the intervention must be guaranteed in a scientific study. This reviewer considers it relevant that this study has sufficient information on the type of intervention performed. It is not enough with this sentence: 

line 134-136: ‘Therapeutic education was based on PNE and CST, whilst therapeutic exercise was composed of strengthening, balance and stability exercises, and divided into core, hip, knee, and global exercises’.

Readers should not have to consult another previous study by the authors to understand what has been done in the present study.

Authors: We thank the reviewer for highlighting the importance of ensuring that the intervention is thoroughly described to guarantee reproducibility and provide sufficient information for readers to understand the methodology. Following the reviewer’s suggestion, we have expanded the description of the therapeutic exercise and orthopedic manual therapy components of the intervention, as follows:

“The intervention consisted of therapeutic education, supervised therapeutic exercise, and orthopedic manual therapy. Therapeutic education was based on PNE and CST, as previously described. Patients in this group received more intensive supervision during exercise training, with increased monitoring of exercise compliance and progression over an 8-week period. The therapeutic exercise program targeted strengthening, balance, and stability, and was organized into exercises focused on the core, hip, knee, and global regions. Core exercises included selective abdominal activation, kneeling abdominal planks, and kneeling side planks. For the hip and pelvic regions, participants performed supine pelvic lifts, one-legged pelvic lifts, lateral hip abductions, and standing hip abductions using elastic bands. Knee-focused exercises included isometric quadriceps contractions, knee extensions with elastic bands, weighted straight leg lifts, half squats with wall support, and alternating leg lunges. Finally, global exercises included one-legged standing, weighted step-ups, slow step-downs, transitioning from a supine to sitting position, and diaphragmatic breathing as a relaxation technique. Orthopedic manual therapy was another key component of this intervention, aimed at alleviating pain and improving mobility. Specific techniques were applied based on the identification of dysfunctions during the intervention period. If participants exhibited a loss of knee range of motion, whether in extension or flexion, mobilizations throughout the range of motion, end-range mobilizations, and mobilizations with movement were performed. Patellar mobility was assessed and mobilized as needed. For muscular tension, dynamic soft tissue mobilizations were employed, while nerve mobilizations and knee tractions were routinely used to modulate pain.”

We hope this expanded explanation meets the reviewer’s expectations and ensures that the intervention is comprehensively described, making the study fully understandable without requiring consultation of previous publications. We greatly appreciate the reviewer’s valuable feedback, which has allowed us to enhance the quality and clarity of the manuscript.